# LoRA-Composer: Leveraging Low-Rank Adaptation for Multi-Concept Customization in Training-Free Diffusion Models

## Abstract

Customization generation techniques have significantly advanced the synthesis of specific concepts across varied contexts. Multi-concept customization emerges as the challenging task within this domain. Existing approaches often rely on training a fusion matrix of multiple Low-Rank Adaptations (LoRAs) to merge various concepts into a single image. However, we identify this straightforward method faces two major challenges: 1) concept confusion, where the model struggles to preserve distinct individual characteristics, and 2) concept vanishing, where the model fails to generate the intended subjects. To address these issues, we introduce LoRA-Composer, a training-free framework designed for seamlessly integrating multiple LoRAs, thereby enhancing the harmony among different concepts within generated images. LoRA-Composer addresses concept vanishing through concept injection constraints, enhancing concept visibility via an expanded cross-attention mechanism. To combat concept confusion, concept isolation constraints are introduced, refining the self-attention computation. Furthermore, latent re-initialization is proposed to effectively stimulate concept-specific latent within designated regions. Our extensive testing showcases a notable enhancement in LoRA-Composer's performance compared to standard baselines, especially when eliminating the image-based conditions like canny edge or pose estimations.

## 1 Introduction

Diffusion models (Rombach et al., 2021) have significantly advanced the field of image generation, particularly in creating images that adhere to user-specific concepts. The progress made in customization models (Gu et al., 2023; Kumari et al., 2022; Shah et al., 2023; Ruiz et al., 2022; Yang et al., 2023; Chen et al., 2023) play an important role in enriching the landscape of image synthesis. As technologies for single concept customization evolve, users are presented with various methods to personalize content, ranging from fine-tuning U-Net (Ruiz et al., 2022; Kumari et al., 2022), modifying text embeddings (Gal et al., 2022; Liu et al., 2023c), to leveraging Low-Rank Adaptations (LoRA) (Hu et al., 2021). LoRA is a versatile, plug-and-play module that enables users to customize their models to generate diverse and lifelike personal images. Its adaptability and accuracy in image generation have established LoRA as a preferred method for customization tasks.

While LoRA excels in single-concept customization, its application to emerging multi-concept customization presents challenges. Recent developments have explored the integration of multiple-concept LoRAs to infuse images with diverse concepts via fusion tuning (Gu et al., 2023; Wang et al., 2023). However, as illustrated in Fig. 1, these integration strategies often necessitate a variety of conditions, including textual and image-based inputs (Zhang et al., 2023) (such as human pose and canny edge), constraining variation and flexibility. Furthermore, in the process of combining multiple LoRAs, prior research (Gu et al., 2023; Wang et al., 2023; Smith et al., 2023) focused on training a fusion ratio matrix, which aims to optimally weigh individual LoRAs. However, adjusting LoRA weights in this manner can exacerbate two problems: 1) concept vanishing, where the concept fails to be injected into the figure; and 2) concept confusion, where the model struggles to associate attributes with subjects or fails to capture distinct concept characteristics. Examples illustrating the issues are displayed in the top row of Fig. 1, showcasing outputs from the representative method Mix-of-Show (Gu et al., 2023). The left column shows a clear case of concept vanishing, where the model fails to generate one of the dogs (within the red box). The right column highlights issues with incorrect attribute binding, such as the dog's color being mistaken (within the blue box).

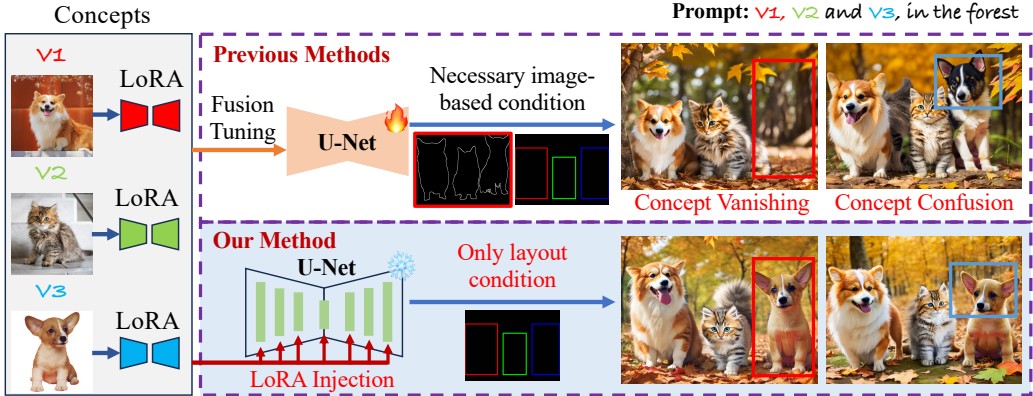

Figure 1: Our method distinguishes itself from Mix-of-Show (Gu et al., 2023) by eliminating the image-based conditions and the requirement to train a LoRA fusion matrix. Furthermore, we highlight the limitations of Mix-of-Show through the demonstration of failure cases. In the top row, we illustrate two key issues: concept vanishing, marked by the absence of intended concepts in the image, and concept confusion, where the model mistakenly merges and confuses distinct concepts.

To overcome the challenges, we introduce LoRA-Composer, a training-free framework that enables the synthesis of images with multiple concept LoRAs, utilizing textual and layout cues. LoRA-Composer encompasses three principal components: concept injection constraints, concept isolation constraints, and latent re-initialization. The concept injection constraints introduce a novel cross-attention mechanism, consisting of 1) region-aware LoRA injection, which injects concept-specific LoRA features into designated regions through cross-attention, facilitating the seamless integration of multiple LoRAs without the need for fusion fine-tuning. 2) concept enhancement constraints, which guide the refinement of latents to accentuate concepts in user-specified regions. These strategies help the model focus on areas designated for concept insertion, effectively mitigating the issue of concept vanishing. The concept isolation constraints address the issue of concept fusion by restricting self-attention, guaranteeing that each concept maintains its unique characteristics. Traditional single-concept LoRAs are typically trained without layout conditions, making them restrictive for localized generation. To tack this, we propose re-initializing the latent vector to establish a more accurate prior, directing the model's focus on specific areas of the image.

We rigorously test our LoRA-Composer across a broad spectrum of multi-concept customization scenarios, including categories such as animals, characters, and scenic backgrounds. Our approach displayed a strong performance compared to existing benchmarks through comprehensive qualitative and quantitative assessments. In summary, our contributions are as follows:

- We propose a training-free model for integrating multiple LoRAs called LoRA-Composer. It requires only easily accessible conditions: layout and textual prompts. This approach simplifies the process of multi-concept customization, enhancing convenience.

- To tackle concept vanishing and confusion, we implement concept injection constraints and concept isolation constraints. These strategies enhance the attention mechanism in U-Net, enabling the model to concentrate on the characteristics of individual concepts and prevent interference from the background or other concepts.

- We propose latent re-initialization to obtain a better prior enhancing the model's capability to focus on specific image sections.

- Our extensive evaluations reveal that our method exceeds baseline performance, particularly in scenarios that eliminate image-based conditions.

## 2 RELATED WORK

### 2.1 CONTROLLABLE IMAGE GENERATION

Diffusion models (Ho et al., 2020; Sohl-Dickstein et al., 2015) trained on large-scale text-to-image datasets, like DALLE-2 (Ramesh et al., 2022), Imagen (Saharia et al., 2022), and Stable Diffu-

sion (Rombach et al., 2021), SDXL (Podell et al., 2023) can produce text-aligned and diverse images in unprecedented high quality. To further support image generation from fine-grained spatial conditions, like sketches, human keypoints, semantic maps, *etc.*, ControlNet (Zhang et al., 2023) finetunes a trainable copy of the pre-trained U-Net and connects the new layers and original U-Net weights with zero convolutions. A similar work, T2I-Adaptor (Mou et al., 2023), finetunes lightweight adaptors for conditional generation from spatial conditions. Differently, GLIGEN (Li et al., 2023b) considers controllable generation with sparse box layout conditions and injects a gated self-attention for fine-tuning. Recent works (Xie et al., 2023; Phung et al., 2023) seek to explore test-time optimization for zero-shot controllable generation. For example, both BoxDiff (Xie et al., 2023) and Attention Refocusing (Phung et al., 2023) achieve zero-shot layout conditioned generation, by maximizing the attention weights between the features inside the box and its corresponding text description, while discouraging the latent features outside the box from attending to the text.

## 2.2 MULTI-CONCEPT CUSTOMIZATION

Concept customization aims at generating concepts specified by a few input images. While significant progress has been made in generating a single custom concept (Ruiz et al., 2022; Gal et al., 2022; Tewel et al., 2023; Shi et al., 2023; Wei et al., 2023; Han et al., 2023; Nichol & Dhariwal, 2021; Li et al., 2023a; Ruiz et al., 2023), the customized generation of multiple concepts remains challenging. A pioneer work, Custom Diffusion (Kumari et al., 2022), jointly finetunes multiple concept images for customization. Cones series (Liu et al., 2023b;d) finds concept-related neurons in pre-trained diffusion models for multi-concept customization. To accelerate the customized generation, FastComposer (Xiao et al., 2023) finetunes diffusion model on massive data to take subject embedding as input and generate the composed image of multiple concepts. Similarly, Paint-by-Example (Yang et al., 2023) and AnyDoor (Chen et al., 2023) are trained on a significant amount of images and can achieve multi-concept generation through image inpainting. Considering the widespread utilization of LoRA for customization , several recent works (Gu et al., 2023; Wang et al., 2023; Shah et al., 2023; Zhong et al., 2024) seek to achieve multi-concept customization by combining multiple LoRA weights of individual concepts. For example, Mix-of-show (Gu et al., 2023) proposes gradient fusion to train a composed LoRA weight that mimics the prediction of individual LoRAs. It further leverages T2I-Adaptor (Mou et al., 2023) and sketches or key points for final generation. A concurrent work (Zhong et al., 2024) proposes two variants, namely LoRA Swich and LoRA Composite, to realize LoRA merge during decoding. The former uses multiple LoRA weights sequentially, while the latter averages the latent obtained from different LoRA weights. However, they focus on the combination of a single foreground and a single background LoRA weights. By contrast, we tackle the more challenging task of customizing multiple foreground characters, facing issues of concept confusion and vanishing.

Probably the most similar work to ours is Mix-of-Show, but we emphasize the following differences. Firstly, Mix-of-Show requires repeated gradient fusion training for each combination of multiple concepts, while we achieve this on the fly without retraining the LoRA weight. Secondly, Mix-of-Show requires additional image-based conditions, like sketches and keypoints, as input for high-quality image generation, which could be difficult to obtain.

## 3 METHOD

In this section, we introduce our innovative LoRA-Composer approach in Sec. 3.1, with the pipeline depicted in Fig. 2(a). The key point is augmenting the scalability of LoRA through the utilization of the LoRA-Composer Block, as illustrated in Fig. 2(b). We then delve into the specifics of the two primary components of the LoRA-Composer Block, outlined in Sec. 3.2 and Sec. 3.3, respectively. Finally, in Sec. 3.4 we discuss the implementation of latent re-initialization to achieve a more refined layout generation prior. We emphasize integrating LoRAs to enable multi-concept customization within a single image, aiming for a solution that is both more flexible and scalable.

## 3.1 LORA-COMPOSER PIPELINE OVERVIEW

As shown in Fig. 2(a), LoRA-Composer utilizes a standard LoRA approach for subject registration, facilitating seamless integration of diverse subjects without requiring training for LoRA fusion. Additionally, to further refine the model's capability in managing multiple conditions simultaneously, we provide the option to incorporate image-based conditions, using T2I-Adapter (Mou et al., 2023).

Figure 2: (a) LoRA-Composer utilizes textual, layout, and image-based conditions (optional) to integrate multiple LoRAs. (b) Modifications to the U-Net in LoRA-Composer Block include concept isolation in self-attention and concept injection in cross-attention. At timestep $t$, $z_t$ is first refined via $\mathcal{L}$ to ensure appearance consistency and prevent feature leakage, followed by the denoising process.

Our primary contribution is the introduction of the LoRA-Composer Block, depicted in Fig. 2(b). In this innovation, we have re-designed the attention block within the U-Net architecture. Specifically, in the cross-attention layers, we implement concept injection constraints designed to counteract concept vanishing. Concurrently, within the self-attention layers, we introduce concept isolation constraints to effectively segregate different concepts, ensuring their distinctiveness. These strategies enable the refinement of the latent space into an image customized according to user preferences, utilizing both self-attention and cross-attention maps to direct the denoise process effectively.

## 3.2 CONCEPT INJECTION CONSTRAINTS

Simply using text prompts to specify desired concepts may result in missing concepts in the basic Stable Diffusion (Chefer et al., 2023). Although spatial attention guidance methods like BoxD-iff (Xie et al., 2023), Attend-and-Excite (Chefer et al., 2023), and Local control (Zhao et al., 2023) can mitigate the issue of missing objects in multi-concept generation, they fall short in precisely represent user-defined concepts. Additionally, Mix-of-Show (Gu et al., 2023) involves optimizing a combination of lesser LoRA weights to preserve the characteristics of the concepts within the pre-trained model. However, this can diminish LoRA's capability to represent conceptual features, resulting in diminished concepts, as illustrated in Fig. 1. To tackle these challenges, we introduce the concept injection constraints, which is comprised of two key components: region-aware LoRA injection and concept enhancement constraints.

**Region-Aware LoRA Injection.** Our approach directly injects each LoRA through region-aware LoRA injection, thereby avoiding the issue of missing concepts caused by the fusion of LoRAs. As illustrated in Fig. 3(a), upon receiving a layout condition, as shown in Fig. 3(b), we extract the queries, keys, and values in the pre-defined layout $M_i$.

$$Q_i = M_i \odot W_0^Q(z), K_i = W_i^K(\tau_i(P^i)), V_i = W_i^V(\tau_i(P^i)), \tag{1}$$

where $i \in \{0, 1, ..., N\}$, and $N$ is number of LoRAs. The pre-trained CLIP text encoder (Radford et al., 2021) combined with LoRA is represented by $\tau_i$. As depicted in Fig. 2(b), for $i \neq 0$, $i$ and $P^i$ correspond to the indices of foreground concept LoRAs and their associated local prompts, respectively. When $i = 0$, these symbols refer to the background concept LoRA and the global prompt that describes the background. The symbol $\odot$ represents the Hadamard product. The $W^Q, W^K$, and $W^V$ stand for the projection matrices within the cross-attention modules of U-Net blocks which combined the concept LoRA. After replicating this process for each concept, we then update the region's hidden state through the cross-attention mechanism as follows:

$$h_i = \text{softmax}\left(\frac{Q_i(K_i)^T}{\sqrt{d}}\right) V_i, \tag{2}$$

Figure 3: Modules of LoRA-Composer Block: (a) region-aware LoRA injection, (b) layout condition, (c) concept region mask, self-attention in the gray area is not calculated.

where the $d$ represents the dimension of queries and keys, this injection approach ensures a comprehensive integration of both background and foreground concepts, enhancing the model's ability to accurately reflect user-specified concepts within the generated images.

**Concept Enhancement Constraints.** Region-Aware LoRA Injection does not sufficiently stimulate all inherent abilities of each LoRA, consequently resulting in the loss of detailed characteristics (see Fig. 6(e)). Previous methods (Xie et al., 2023; Chefer et al., 2023) have proposed enhancing activation within the cross-attention mechanism in specific regions to improve the ability of Stable Diffusion (Rombach et al., 2021) to represent concepts effectively. However, these approaches are unsuitable for multi-concept customization tasks, as object generation tends to occur near the edges of the layout box (see Appendix Fig. 9(a)) or fails to fill it adequately (see Appendix Fig. 9(b)). To address the first issue, we introduce a Gaussian weight into the cross-attention map to restrain activation in the edge region. For the second issue, we design a loss function, $\mathcal{L}_f$, to ensure large activation values are spread over the box area as evenly as possible.

$$\mathcal{L}_c = \sum_{i=1}^{N}(1 - \frac{1}{S}\sum_{j\in E}\mathbf{topk}(A_i^j \odot M_i \odot G, S)), \tag{3}$$

where $\mathbf{topk}(\cdot, S)$ indicates the selection of $S$ elements with the highest response in input and $G$ means standard Gaussian distribution. We define the token that stimulates the concept in LoRA as a concept token (for more details, see Appendix A). The symbol $E$ represents the collection of these concept tokens. For the foreground concept $i$, the cross-attention map corresponding to the $j$-th token is represented by $A_i^j$. As for the $\mathcal{L}_f$, we squeeze cross-attention maps on the w-axis and h-axis via the max operation as below:

$$a_i^j(w) = \mathbf{max}_{h=1,2,...,H}\{M_i \odot A_i^j(w,h)\}, \tag{4}$$

$$a_i^j(h) = \mathbf{max}_{w=1,2,...,W}\{M_i \odot A_i^j(w,h)\}, \tag{5}$$

where the variables $W$ and $H$ denote the width and height of the cross-attention map $A_i^j$. Then we compute the L1 loss in each axis as follows:

$$\mathcal{L}_f = \frac{1}{L}\sum_{i=1}^{N}\sum_{j\in E}(\mathbf{1} - \{a_i^j(w), a_i^j(h)\}), \tag{6}$$

where $\{\cdot, \cdot\}$ denotes the concatenation followed by flattening and $\mathbf{1}$ represents a vector of ones. The term $L$ is defined as the length of $\mathbf{1}$.

## 3.3 CONCEPT ISOLATION CONSTRAINTS

While the concept injection constraints effectively guarantee that objects will be placed within user-specified regions, they cannot prevent the potential overlap or infection of customized concepts within these areas (see Fig. 6(d)). To preserve the integrity and distinctiveness of each concept within its designated region, we introduce concept isolation constraints. This approach is divided into two main components: concept region mask and region perceptual restriction. Both elements are integrated within the self-attention of the U-Net block, ensuring that each concept remains isolated and unaffected by others, thereby maintaining the purity of concepts in target regions.

**Concept Region Mask.** The self-attention mechanism creates connections among all query elements, essential for maintaining the distinct characteristics of each concept. To preserve the distinctiveness of each concept, we adopt the concept region mask strategy, guided by a given layout condition as depicted in Fig. 3(b). This design limits the interaction between queries within a specific concept region and those in other concept regions, as demonstrated in Fig. 3(c). Thus, it ensures the preservation of each concept's characteristics, free from the influence of neighboring concepts.

**Region Perceptual Restriction.** Due to down-sampling and residual convolution operations in the U-Net framework, concept features might spread into the elements designated for background areas, as highlighted by the yellow square in Fig. 3(b). To mitigate the risk of concept feature leakage into unintended regions, we employ region perceptual restriction, aimed at minimizing interaction between queries of the foreground and background areas. This technique ensures that each concept remains distinct and unaffected by the features of the background feature, thereby preserving the uniqueness and integrity of each concept within the synthesized image. This formulated as

$$\mathcal{L}_r = \frac{1}{S} \sum_{i=1}^{N} \textbf{topk}(\bar{A}[M_i, \mathbf{1} - M_i], S), \tag{7}$$

where the $\bar{A}[M_i, \mathbf{1} - M_i]$ denotes the self-attention map derived by slicing across the channel dimension to extract attention scores between foreground and background pixels.

At each timestep, overall constraints loss are formulated as:

$$\mathcal{L} = \mathcal{L}_c + \alpha\mathcal{L}_f + \beta\mathcal{L}_r, \tag{8}$$

where $\alpha$ and $\beta$ represent weighting coefficients. Using the constraints loss $\mathcal{L}$, the current latent $z_t$ can be updated with a step size of $\phi_t$ as follow:

$$z'_t \leftarrow z_t - \phi_t \cdot \nabla\mathcal{L}. \tag{9}$$

Following BoxDiff (Xie et al., 2023), the step size $\phi_t$ decays linearly with each timestep. By incorporating the previously mentioned constraints, $z'_t$ is directed at each timestep to foster the generation of customized concepts within designated locations, while preventing the leakage of concept features into areas associated with other concepts. Subsequently, $z'_t$ is utilized as the input for the U-Net for the ensuing inference step $z'_t \xrightarrow{U-Net} z_{t-1}$. This strategic guidance ensures the precise synthesis of target concepts within the user-specified layout regions.

### 3.4 LATENT RE-INITIALIZATION

We discovered that traditional LoRA is not ideally suited for generating specific local areas, because it is trained without control in location. This discrepancy can result in imprecise locations for concept generation (see Fig. 6(c)). To address this issue, we propose re-initializing the latent space to better accommodate the integration of concept-specific LoRAs.

Our approach aims to identify the position within the latent space where the object is likely to appear and then align this position with the specified layout. Specifically, before the denoising phase, we initialize the latent space $z_t$ with Gaussian noise and apply the LoRA-Composer process for a one-step update using Eq. (9). Afterward, a cross-attention map is generated based on the region latent query $Q_i$ and the textual embedding $K_i$ for each local prompt. First, we compute each candidate area with the same shape as the layout. Then, we replace the layout area $z_t[M_i]$ with the latent area corresponding to the highest scoring area among the candidate areas. Finally, the latent is normalized to a standard Gaussian distribution. The aforementioned candidate areas can be expressed as:

$$\hat{A}_i = \{\Phi(A_i, [x, y], \mathbb{W}, \mathbb{H})\}, \tag{10}$$

where $x \in \{0, 1 \ldots w\}, y \in \{0, 1 \ldots h\}$. The $w$ and $h$ denote the width and height of the cross-attention map $A_i$. The function $\Phi(\cdot, [x, y], \mathbb{W}, \mathbb{H})$ refers to cropping the attention map to a shape of $\mathbb{W}, \mathbb{H}$ with the top-left coordinate at $[x, y]$. The variables $\mathbb{W}$ and $\mathbb{H}$ represent the width and height of the layout box.

## 4 EXPERIMENTS

### 4.1 EXPERIMENTAL SETUP

**Datasets**. For a thorough evaluation of LoRA-Composer, follow Mix-of-Show (Gu et al., 2023), we compile a dataset featuring characters, animals, and backgrounds in both realistic and anime styles

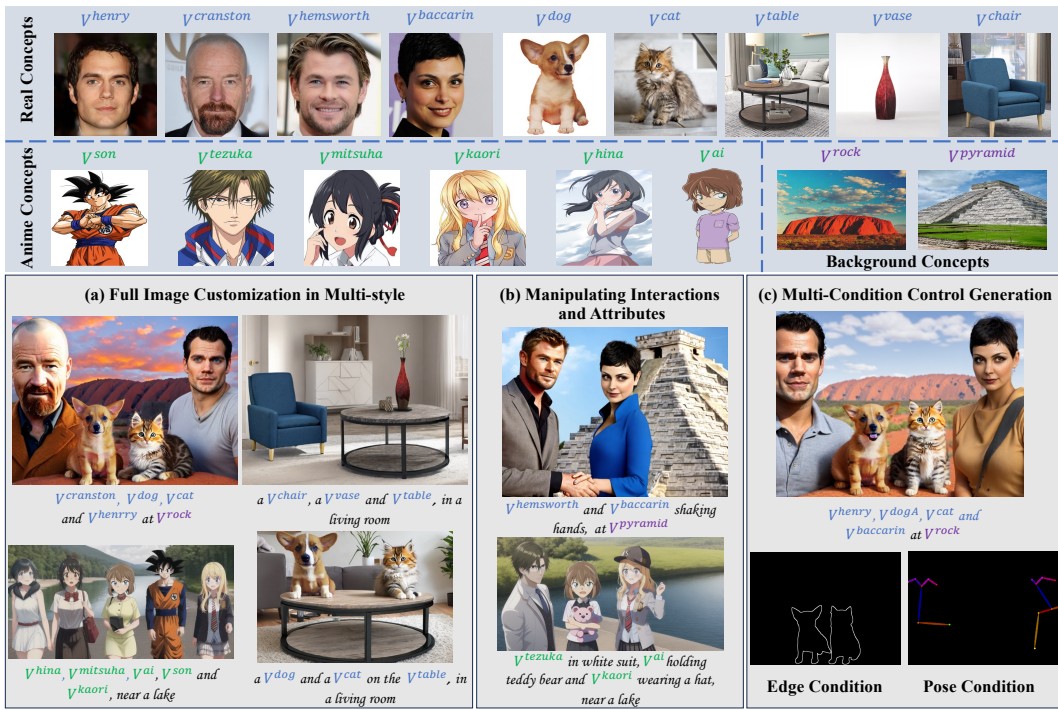

Figure 4: Three highlights of LoRA-Composer, a) full image customization in multi-style; b) manipulating interactions and attributes; c) multi-condition control generation.

for a comprehensive evaluation. The dataset encompasses a total of 15 customized subjects. Please refer to Appendix Fig. 7 for more details.

**Evaluation metrics**. Following prior methods (Liu et al., 2023c; Gu et al., 2023; Kumari et al., 2022), we utilize two CLIP metrics (Radford et al., 2021): (1) Image similarity (indicated by I): Measures the visual resemblance between generated images and target subjects in CLIP embeddings. (2) Textual Similarity (indicated by T): Assesses the alignment between generated text and target text descriptions. Additionally, to evaluate the precision of concept localization we introduce the mIoU metric. For detailed information, please refer to Appendix B.

**Baseline**. We compare our approach against four leading competitors in the field. **Cones2** (Liu et al., 2023c) leverages text embeddings to support arbitrary combinations of concepts. **Mix-of-Show** (Gu et al., 2023) employs gradient fusion to integrate multiple concepts into a base model. Both **Anydoor** (Chen et al., 2023) and **Paint by Example** (Yang et al., 2023) facilitate multi-concept generation by utilizing networks trained specifically for inpainting tasks.

## 4.2 VISUALIZATION RESULTS

Our broad customization capability is illustrated in Fig. 4(a), showcasing our approach's versatility in adapting to a wide range of styles, from anime to realistic. Additionally, our method enables precise manipulation of interactions and attributes, such as shaking hands, wearing hats, and holding teddy bears in the picture, directly through textual prompts. This capability is showcased in Fig. 4(b). Moreover, our framework is designed for flexibility, capable of generating images under multiple conditions. It adeptly integrates specific constraints such as edge detection or pose estimation to guide the image synthesis process. This capacity for accommodating additional image-based conditions, as detailed in Fig. 4(c), highlights the adaptability of our approach in meeting varied and complex generation requirements. More visualization results are shown in Fig. 12 in the Appendix.

## 4.3 QUANTITATIVE RESULTS

As detailed in Tab. 1, our LoRA-Composer surpasses prior methods in image similarity, showcasing its effectiveness across both anime and realistic styles. Conversely, inpainting-based methods such as Anydoor and Paint by Example exhibit higher text similarity and mIoU. This is because these methods specialize in inserting subjects into user-defined locations through reference images,

| Method | Anime-I | Anime-T | Real-I | Real-T | Mean-I | Mean-T | Real-mIoU |
|---|---|---|---|---|---|---|---|
| Cones2 (Liu et al., 2023c) | 0.5940 | 0.5691 | 0.5106 | 0.5948 | 0.5523 | 0.5820 | 0.226 |
| Mix-of-Show (Gu et al., 2023) | 0.6296 | 0.5741 | 0.6015 | 0.5977 | 0.6156 | 0.5859 | 0.347 |
| Anydoor (Chen et al., 2023) | - | - | 0.6398 | **0.6379** | 0.6398 | **0.6379** | 0.537 |
| Paint by Example (Yang et al., 2023) | - | - | 0.6370 | 0.6286 | 0.6370 | 0.6286 | **0.612** |
| **LoRA-Composer** | **0.8219** | **0.5945** | **0.7115** | 0.6284 | **0.7667** | 0.6114 | 0.571 |
| Mix-of-Show* (Gu et al., 2023) | 0.8238 | **0.6067** | 0.6536 | 0.6229 | 0.7387 | 0.6148 | 0.577 |
| **LoRA-Composer*** | **0.8320** | 0.5981 | **0.6911** | **0.6323** | 0.7615 | **0.6152** | **0.616** |

Table 1: Quantitative comparison of LoRA-Composer with baselines in generating anime and realistic style concepts. **T** refers to textual similarity, **I** refers to image similarity, with an asterisk * indicating the use of image-based conditions. The highest scores are marked in **bold**.

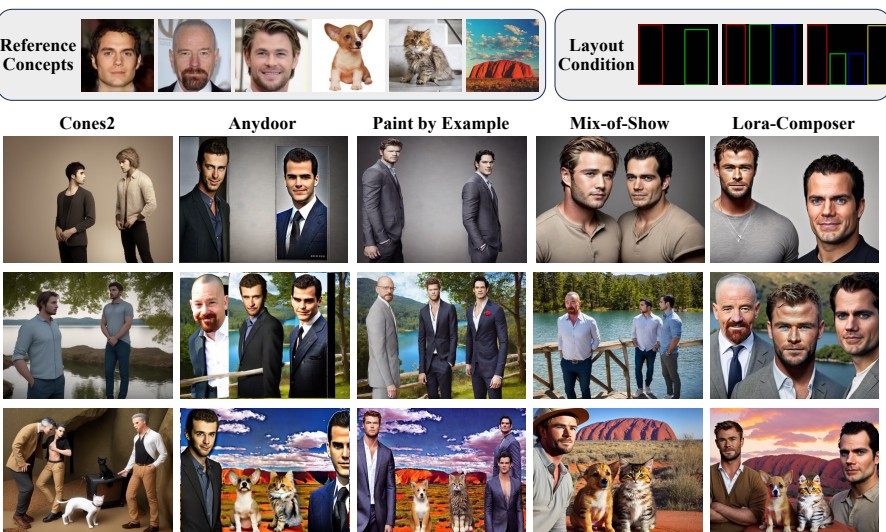

Figure 5: Qualitative comparison with baselines. For each case, we use the same seeds.

focusing more on aligning with textual descriptions. However, they struggle to maintain the characteristics of the concept. Our method significantly enhances image quality, achieving the highest scores in image similarity.

To ensure fairness in our comparison, we established two settings: one without using image-based conditions and another incorporating them (indicated by *). Our method consistently outperforms in both scenarios. We observed that image-based conditions play a crucial role in Mix-of-Show. Without these conditions, it faces severe drops in both image and textual similarity. In contrast, LoRA-Composer exhibits enhanced robustness and gains further advantage from image-based conditions, offering increased convenience to users.

### 4.4 QUALITATIVE COMPARISON

LoRA-Composer is evaluated without using image-based conditions across four benchmarks in multi-concept customization scenarios. The results are displayed in Fig. 5. It can be seen that Cones2 (Liu et al., 2023c), Anydoor (Chen et al., 2023), and Paint by Example (Yang et al., 2023) face concept confusion, unable to clearly capture concept characteristics. These approaches also lead to disproportionate foreground elements and unnatural integration of foreground and background. Mix-of-Show (Gu et al., 2023) also suffers from concept confusion, as illustrated in the first row, where the man on the left loses his distinguishing features. A similar issue occurs in the second row. In the third row, one individual disappears entirely.

Differently, our method successfully synthesizes images that accurately incorporate all subjects with their correct characteristics, showcasing enhanced performance in multi-concept synthesis and attribute accuracy. Additional qualitative comparison results and results using image-based condition guidance are provided Fig. 8 and Fig. 10 in the Appendix.

| CE | CI | LR | Anime-I | Anime-T | Real-I | Real-T | Mean-I | Mean-T | Real-mIoU |
|----|----|----|---------|---------|--------|--------|--------|--------|-----------|
| ✓ | ✓ | ✓ | 0.8131 | 0.5948 | 0.7106 | 0.6336 | 0.7618 | 0.6142 | 0.455 |
| ✓ | ✓ |   | 0.8024 | 0.5923 | 0.7105 | 0.6344 | 0.7565 | 0.6134 | 0.443 |
| ✓ |   |   | 0.7957 | 0.5899 | 0.6271 | 0.6096 | 0.7114 | 0.5997 | 0.343 |
|   |   |   | 0.6597 | 0.5725 | 0.6067 | 0.6041 | 0.6332 | 0.5883 | 0.287 |

Table 2: Ablation studies on various components. **"LR"** stands for latent re-initialization, **"CI"** denotes concept isolation constraints, and **"CE"** signifies concept enhancement constraints within concept injection constraints.

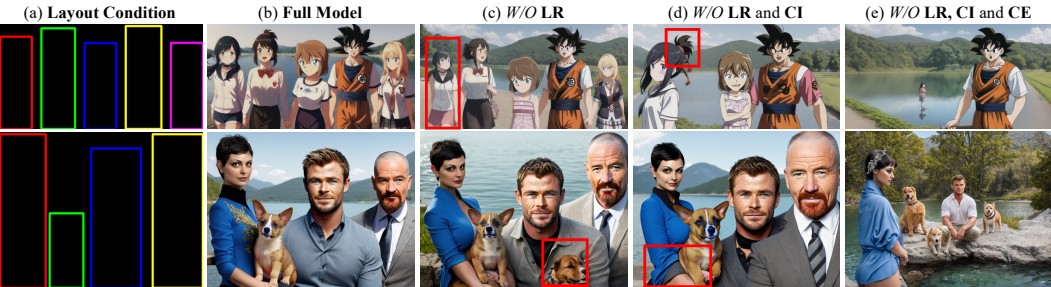

Figure 6: Visualized results from ablation study on individual components. **"LR"** stands for latent re-initialization, **"CI"** denotes concept isolation constraints, and **"CE"** signifies concept enhancement constraints within concept injection constraints.

## 4.5 ABLATION STUDY

To demonstrate the efficacy of each component in our method, we choose a challenging scenario with five and four concepts for this section. As illustrated in Fig. 6(c), the positions of the anime girl and the dog within the red box diverge from the layout conditions outlined in Fig. 6(a). This discrepancy serves as evidence that omitting latent re-initialization (LR) hampers the precise placement of concepts, mainly because of the lack of spatial priors. Subsequently, as shown in Fig. 6(d), removing the concept isolation constraints (CI) results in the blending of concept characteristics. This results in observable issues such as the confusion in the anime girl's haircut, and the distortion of the woman's arm. Without CI, concepts begin to overlap and influence each other, resulting in a disruption of harmony and coherence in the overall image composition. Finally, as shown in Fig. 6(e), eliminating the concept enhancement constraints (CE) results in the disappearance of concepts. However, thanks to the presence of region-aware LoRA injection, the model retains the capability to insert concepts, though with diminished precision in their placement and representation. This highlights each element's critical role in achieving precise and harmonious concept integration.

To substantiate our findings as non-coincidental, we conducted a comprehensive quantitative evaluation. As detailed in Tab. 2, our analysis demonstrates the individual and collective impacts of CE, LR, and CI on performance. CE leads to significant improvements across all performance metrics, showcasing its effectiveness in activating concepts, as evidenced by the largest increase in mean image similarity. LR further contributed to these enhancements by refining region-specific priors. CI played a crucial role in preserving the distinctiveness of concept traits and enhancing model robustness. More ablation results are shown in Appendix C.2.

## 5 CONCLUSION

In this paper, we introduce LoRA-Composer, a novel approach designed to seamlessly integrate multiple concepts within a single image. We explore two prevalent issues in multi-concept customization: concept vanishing and concept confusion. To this end, we employ concept injection constraints to combat concept vanishing, while concept isolation constraints alleviate concept confusion. Additionally, we propose latent re-initialization to provide precise region priors. Our experiments highlight the capability of LoRA-Composer to customize entire images, including both background and foreground elements, and to manipulate the interactions and attributes of various concepts through textual prompts. Compared to traditional methods, LoRA-Composer offers enhanced flexibility and usability, allowing users to generate images with fewer conditions and readily accessible LoRA techniques. Furthermore, we demonstrate the method's ability to achieve high-fidelity combinations of multiple concepts, underscoring its practical utility in complex image-generation tasks.

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

APPENDIX

Considering the space limitation of the main text, we provided more results and discussion in this supplementary material, which is organized as follows:

- Section A: a concise overview of diffusion models (Rombach et al., 2021) and ED-LoRA (Gu et al., 2023).
- Section B: implementation details of our approach and the baseline models.
- Section C: more detailed experiments analysis and discussion.
  - Section C.1: our default setting in experiments and ablation study.
  - Section C.2: ablation study on concept enhancement constraints (in Sec. 3.2) and concept isolation constraints (in Sec. 3.3).
  - Section C.3: comparison with Mix-of-Show, under the image-based conditions.
  - Section C.4: assess the human preference between our method and baseline approaches.
  - Section C.5: more visual results of LoRA-Composer.
- Section D: discussion of our method's potential negative society impact.
- Section E: failure cases and discussion.

## A PRELIMINARY

**Diffusion Models** are famous for their capacity to generate high-quality images. Their framework operates in two primary phases: the forward phase, where Gaussian noise is progressively added to an image until it fully conforms to a Gaussian distribution, and the reverse phase, which aims to reconstruct the original image from its noised condition. The reverse phase typically employs a U-Net architecture enhanced with text conditioning, enabling the synthesis of images based on textual descriptions during inference. In this work, we employ Stable Diffusion (Rombach et al., 2021), which distinguishes itself by operating in the latent space rather than directly manipulating image pixels through these phases. This approach involves an autoencoder, with an encoder $\mathcal{E}$ and decoder $D$, trained to serve as a bridge between image pixel space $x$ and latent space $z$, i.e., $D(z) = D(\mathcal{E}(x))$. In each time step $t$, given a textual condition $\tau(P)$ and an image $x$, where $P$ represents the text prompt and $\tau$ denotes the pre-trained CLIP text encoder (Radford et al., 2021). The training objective for Stable Diffusion is to minimize the denoising objective by

$$\mathcal{L}_{sd} = \mathbb{E}_{z \sim \mathcal{E}(x), P, \epsilon \sim \mathcal{N}(0,1), t} \left[ \| \epsilon - \epsilon_\theta(z_t, t, \tau(P)) \|_2^2 \right], \tag{11}$$

where $\epsilon_\theta$ is the denoising U-Net with learnable parameter $\theta$.

**ED-LoRA** (Gu et al., 2023) aims to augment the expressiveness of the embedding by employing a decomposed structure. We use it by default. ED-LoRA implements a layer-wise embedding strategy, following the method described in P+ (Voynov et al., 2023), to forge a multi-faceted representation for the concept token ($V = [V_{rand}, V_{class}]$). This involves adding a random variation ($V_{rand}$) and a class-specific component to the base embedding ($V_{class}$). Furthermore, it integrates a LoRA layer into the linear layers present in all attention modules of the text encoder and U-Net. This integration allows for a flexible adaptation of the model to specific concepts by modifying the linear layers in a low-rank manner, thereby enhancing the model's ability to encode and synthesize images based on textual descriptions with high fidelity. We use it by default in all experiments.

## B IMPLEMENTATION DETAIL

**ED-LoRA Setting.** We chose ED-LoRA due to its strong capability in maintaining concept fidelity. In alignment with the single-concept ED-LoRA tuning guidelines from (Gu et al., 2023), we integrate the LoRA layer into the linear layers of all attention modules within both the text encoder and U-Net, setting a rank ($r$) of 4 for all experiments. For optimization, we employ the Adam optimizer (Kingma & Ba, 2014), utilizing learning rates of 1e-5 for the text encoder and 1e-4 for U-Net tuning. Follow Mix-of-Show (Gu et al., 2023), we use 5-20 images for training each concept.

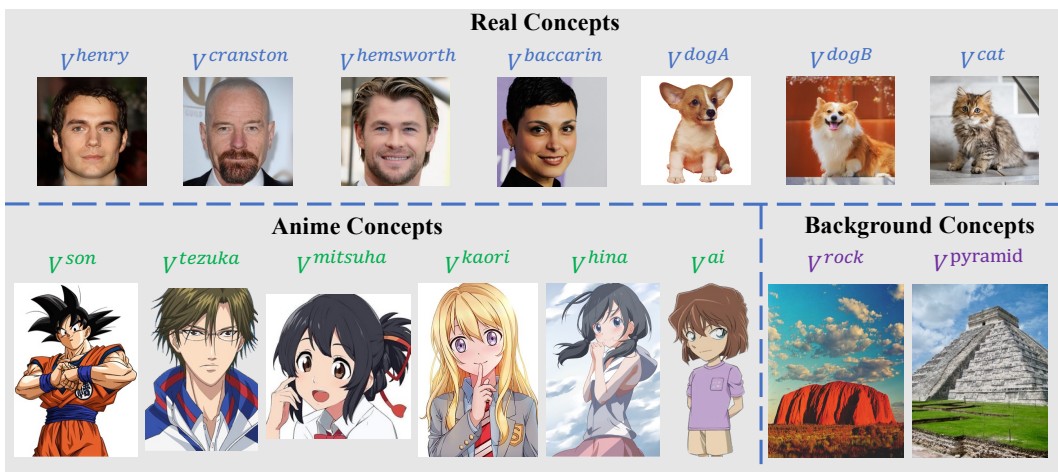

Figure 7: The datasets utilized for our model encompass a diverse range of concepts, including real-world objects, anime characters, and background scenes, totaling 15 distinct concepts.

**Sample Details.** For all experiments and evaluations in this paper, we use the DPM-Solver (Lu et al., 2022), implementing adaptive sampling steps to enhance computational efficiency. Specifically, if the loss (as described in Eq. (9)) ceases to decrease, we stop the process, thereby accelerating the overall procedure. For this loss, the relative coefficients are set as $\alpha = 0.25$ and $\beta = 0.8$.

**Pretrained Models.** Following the approach used by Mix-of-Show (Gu et al., 2023), we select Chilloutmix[1] as the pre-trained model for crafting real-world concept images. For anime concepts, Anything-v4[2] serves as the chosen pre-trained model. To ensure equitable comparisons among different methods, all comparative analyses involving training-based methods, such as Cones2 (Liu et al., 2023c) and Mix-of-Show (Gu et al., 2023), utilize these specified pre-trained models, guaranteeing uniformity in evaluation criteria. For inpainting-based models, specifically Anydoor (Chen et al., 2023) (which refines Stable Diffusion v2.1) and Paint by Example (Yang et al., 2023) (which refines Stable Diffusion 1.4), we adhere to their official models.

**Evaluation metrics.** **(1) Image similarity**: In our multi-concept generation task, we crop each foreground concept within the generated images, while using the entire image for the background. We compute the image similarity for each target concept individually and then calculate the average value. **(2) Textual similarity**: We extract every concept token $V$ in local prompt $P^i$ and substitute it with the concept's class name before computing the textual similarity. The final score is the average of these similarity values. **(3) mIoU metric**: We extract the bounding boxes of the foreground concepts by GroundingDINO (Liu et al., 2023a) and compute the mIoU between these boxes and the layout conditions. We only evaluate this metric in real style, because almost all detection models are trained with realistic datasets.

**Baseline Implementation Detail.** For **Cones2** (Liu et al., 2023c), we utilize the official implementation provided at the repository[3]. The training configurations specified include a batch size of 4, a learning rate of 5e-6, and a total of 4000 training steps. This setup requires approximately 10-15 minutes to execute on a single NVIDIA A100 GPU. To ensure consistency across experiments, we employ the same seed for image generation. Given that **Paint by Example** (Yang et al., 2023)[4] and **Anydoor** (Chen et al., 2023)[5] focus exclusively on real object inpainting, we ensure a fair comparison by limiting the comparison to real-world concepts. Specifically, our approach involves initially generating the background image using our model with the same prompt and seeds, while omitting the foreground prompts. Subsequently, their models are employed to introduce the foreground concepts. For **Mix-of-Show** (Gu et al., 2023), we utilize the same LoRAs for both real-world and anime

---

[1]https://huggingface.co/windwhinny/chilloutmix
[2]https://huggingface.co/xyn-ai/anything-v4.0
[3]https://github.com/ali-vilab/Cones-V2
[4]https://github.com/Fantasy-Studio/Paint-by-Example
[5]https://github.com/ali-vilab/AnyDoor

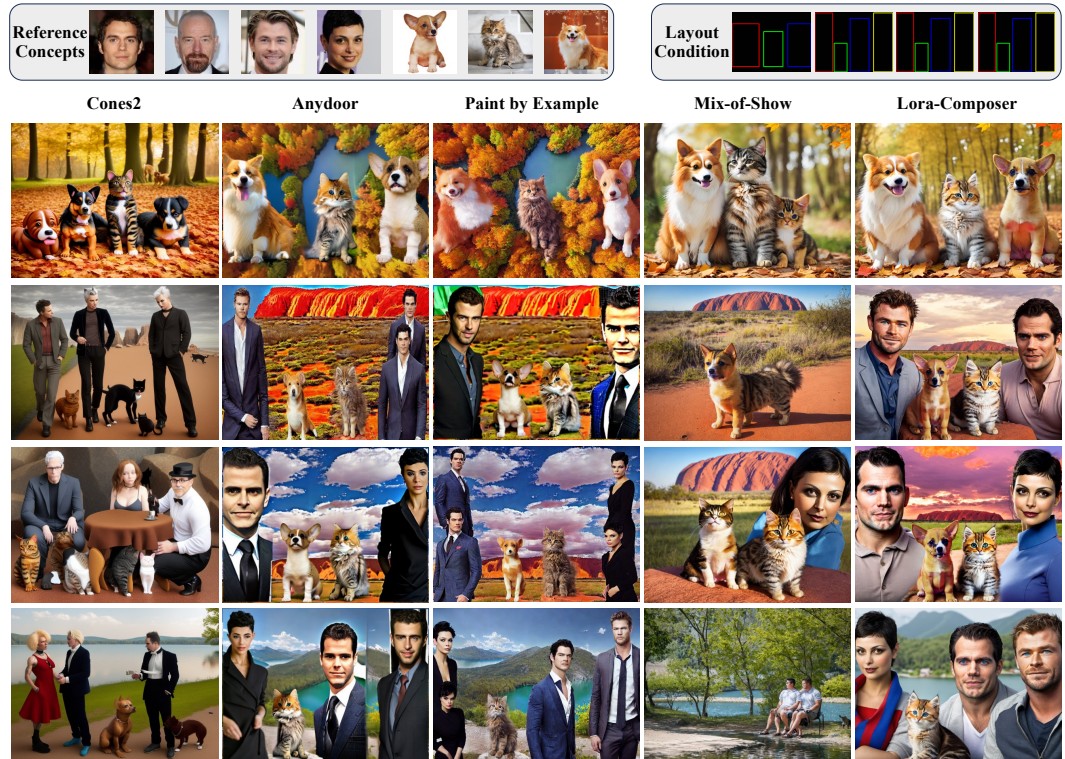

Figure 8: More qualitative comparison with baselines. For each case, we use the same seeds.

concepts. We apply their gradient fusion technique[6] to integrate all of the LoRAs into the base model. Consistency across experiments is ensured by using the same seed for image generation, allowing for a direct comparison of outcomes.

## C ADDITIONAL EXPERIMENTS

### C.1 DEFAULT SETTING IN EXPERIMENTS

We collect a diverse dataset featuring characters, animals, and backgrounds in both realistic and anime styles, encompassing 15 unique subjects (as shown in Fig. 7). To comply with privacy regulations, all real-person concepts we use are sourced from the CelebA-HQ dataset (Karras et al., 2018). To assess our model, we randomly picked three varied settings in two styles, testing combinations of two to five subjects. We produced 50 images for each setting, culminating in $2 \times 3 \times 4 \times 50 = 1200$ images for an extensive performance review.

For our ablation study, we selected three challenge settings within both anime and realistic styles, involving four and five concepts. This approach yielded 600 images, offering a substantial dataset to examine the effects of different model components and settings on our framework's capability.

### C.2 MORE ABLATION STUDY

In the main ablation study (Sec. 4.5), we explored the synergistic effects of combining modules with similar functionalities. Here, we delve deeper with an extensive ablation study on our concept enhancement constraints, which includes Gaussian sample strategy in Eq. (3) and $\mathcal{L}_f$ in Eq. (6)) and concept isolation constraints (incorporating the concept region mask and $\mathcal{L}_r$ in Eq. (7)). These examinations aim to illuminate their contributions to model performance, as illustrated in Fig. 9. Specifically, in the first column, the absence of Gaussian sampling leads to the concepts not being accurately centered within their designated boxes. This lack of precision can even cause anime concepts to appear outside their intended boundaries, resulting in a loss of their unique identity traits.

---

[6]https://github.com/TencentARC/Mix-of-Show/tree/research_branch

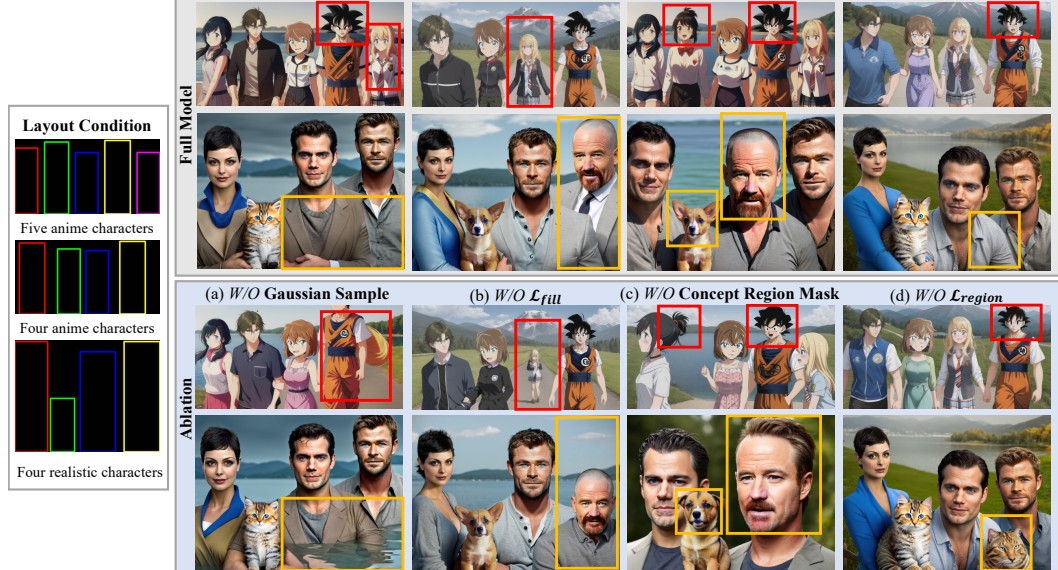

Figure 9: More ablation study on concept enhancement constraints and concept isolation constraints. The upper portion displays outcomes utilizing our full methodology, while the lower portion illustrates results with specific modules omitted, highlighting the significance of each component within our approach. The red boxes and the yellow boxes are used to accentuate the distinctions between the anime style and the real-world style, respectively.

In the second column, without $\mathcal{L}_f$, both anime and realistic figures fail to occupy their designated boxes completely, pointing to a deficiency in fully utilizing the allocated space. In the third column, we observe concept confusion, characterized by the merging of anime haircuts and the loss of distinctive facial traits in realistic figures, which indicates a loss of distinctiveness. This highlights the role of the concept region mask in safeguarding each concept's unique attributes. In the last column, concept features, such as the anime boy's haircut being influenced by another character, and an unintended cat appearing. These issues indicate that there is leakage into unintended areas, due to down-sampling in U-Net. The inclusion of $\mathcal{L}_r$ effectively addresses this problem by minimizing the influence of background elements on foreground concepts. These strategies validate the essential roles played by the concept enhancement and concept isolation constraints in maintaining concept integrity and precision within the generated images, significantly bolstering the model's capability to produce conceptually coherent and visually accurate outputs.

## C.3 MORE COMPARISON

To ensure a fair comparison with Mix-of-Show (Gu et al., 2023), we adopted their default settings, applying the same image-based conditions and using identical random seeds for generating multi-concept images. The comparative results are depicted in Fig. 10, showcasing four unique concept combinations styled in both anime and real-world visuals. Our analysis reveals that while Mix-of-Show struggles with maintaining distinct identity features (as indicated by yellow boxes) and the completeness of the integrated concepts (highlighted by red boxes), our approach successfully overcomes these limitations. Our method produces high-fidelity, coherent images that significantly enhance user satisfaction and improve the perceived quality of the generated content.

## C.4 USER STUDY

To assess the efficacy of our multi-object customization outcomes more accurately, we implemented a user study to capture human preferences. Following the approach utilized in Mix-of-Show, participants evaluated the generated images based on two key metrics: 1) Text-to-Image Alignment: This assesses how well the textual description matches the generated image. 2) Image-to-Image

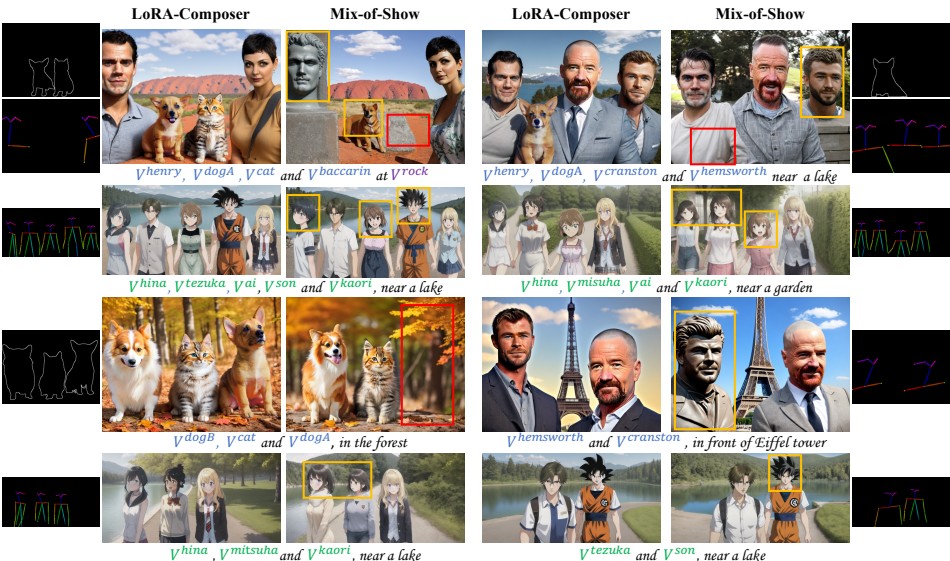

Figure 10: Comparison with Mix-of-Show, where image-based conditions are applied. The yellow box emphasizes the issue of concept confusion, while the red boxes underscore instances of concept vanishing.

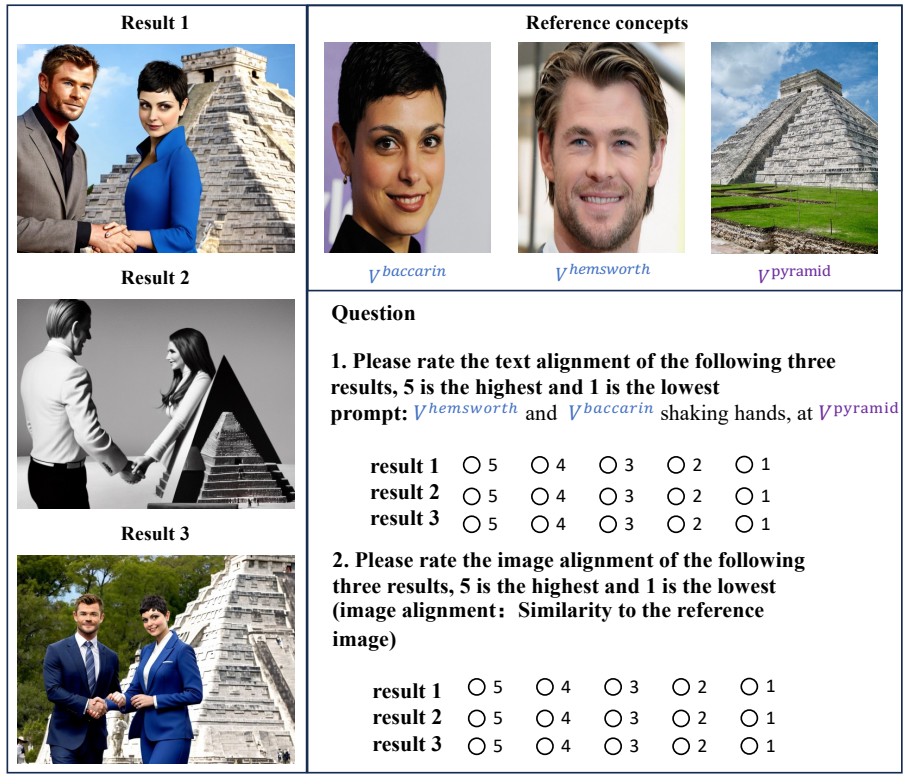

Figure 11: The user study interface that participants used to evaluate generated images on text-to-image alignment and image-to-image alignment.

Alignment: This examines the resemblance between the character in the generated image and the provided character reference image.

As shown in Fig. 11, participants rated each aspect on a scale from 1 to 5, where higher scores denote superior quality. To thoroughly gauge the performance across various multi-object customiza-

| Method | Text-to-Image | Image-to-Image |
|---|---|---|
| Cones2 (Liu et al., 2023c) | 1.99 | 1.25 |
| Mix-of-Show (Gu et al., 2023) | 3.13 | 2.58 |
| Anydoor (Chen et al., 2023) | 2.73 | 2.07 |
| Paint by Example (Yang et al., 2023) | 2.19 | 1.53 |
| **LoRA-Composer** | **4.25** | **4.02** |
| Mix-of-Show* (Gu et al., 2023) | 3.84 | 3.28 |
| **LoRA-Composer*** | **4.23** | **3.78** |

Table 3: User study. The scores reflect user preferences, with higher values indicating better quality. It shows that our approach is favored by users for multi-concept customization, excelling in both image and text alignment. An asterisk * denotes using image-based conditions. The highest scores in each column are marked in **bold**.

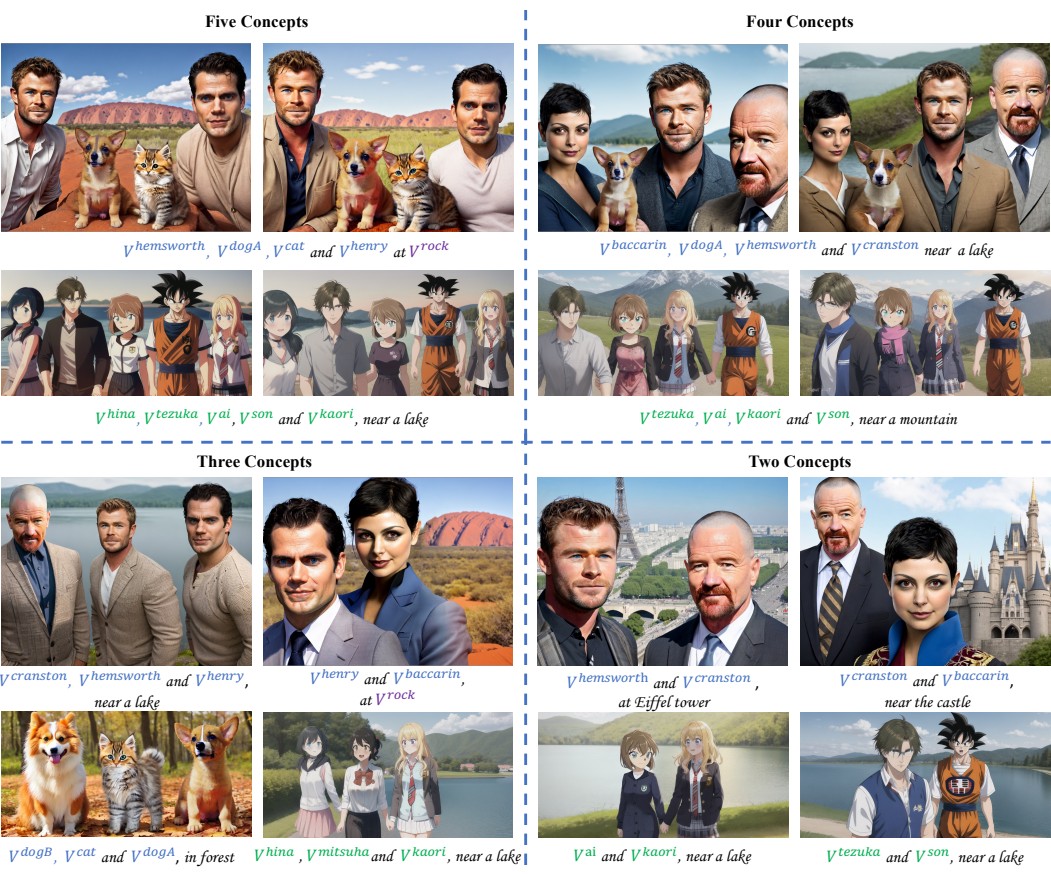

Figure 12: More results of our method in four configurations.

tion scenarios, we included setups involving 2, 3, 4, and 5 customization concepts. The sequence of all image-question pairs was randomized before being presented to 25 different users for evaluation. Each user was tasked with rating a total of 60 questions. The study results are shown in Tab. 3. Across all scenarios, LoRA-Composer emerged as the preferred choice, receiving the highest score of votes. Notably, our method demonstrated significant strengths, especially in scenarios that required eliminating image-based conditions. These outcomes demonstrate the effectiveness of LoRA-Composer in the generation of multi-concept customized images.

## C.5 MORE VISUAL RESULTS

In Fig. 12, we showcase an extended collection of images produced using our method. This display illustrates the superior flexibility and usability of LoRA-Composer, enabling users to create images

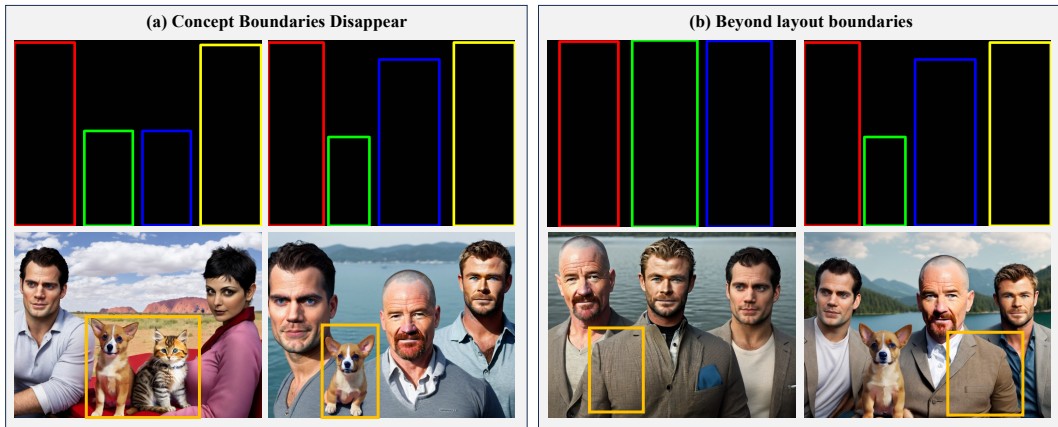

Figure 13: Two limitations of LoRA-Composer.

under few conditions and utilizing easily accessible LoRA techniques. Additionally, our method's capability to seamlessly blend multiple concepts into high-fidelity images showcases its effective application in multi-concept generation tasks.

## D  POTENTIAL NEGATIVE SOCIETY IMPACT

This project is dedicated to offering the community an advanced tool for multi-concept image customization, empowering users to merge various concepts seamlessly to craft complex visuals. Nonetheless, there's a risk that such a powerful framework could be misused by malicious parties to create deceptive interactions with real-world figures, posing potential harm to the public. To counteract these risks, one potential solution is implementing protective measures akin to those proposed in DUAW (Ye et al., 2023), which introduces a universal adversarial watermark. This watermark is designed to interfere with the variational autoencoder's function, thereby hindering the model's ability to be exploited for malicious customization.

## E  LIMITATION AND FUTURE WORK

The first limitation is about disappearing concept boundaries (Fig. 13(a)): This issue arises when the space between concepts is too small, causing potential overlap due to down-sampling. Increasing the spacing between concepts can alleviate this problem.

The second limitation pertains to instances where concepts extend beyond their designated layout boundaries, as shown in Fig. 13(b). Occasionally, foreground elements may spill over their intended borders, a consequence of Stable Diffusion's (Rombach et al., 2021) design, which relies on generalized assumptions to generate outcomes. Adopting a more structured layout strategy could potentially mitigate this issue.

The final limitation pertains to inference efficiency. A slight delay occurs due to the need to load various LoRA checkpoints and perform backward computations to update latent representations. This process takes approximately 20-40 seconds per image on a single NVIDIA A100 GPU.

In future work, we aim to enhance the attention mechanism to overcome existing limitations and optimize the IO process to improve inference efficiency.

