# OpenReview forum: "LoRA-Composer: Leveraging Low-Rank Adaptation for Multi-Concept Customization in Training-Free Diffusion Models"
_ICLR.cc/2025/Conference — ICLR 2025 Conference Withdrawn Submission_

### Official Review · Reviewer_QXPv · 2024-10-21

**Soundness:** 3
**Presentation:** 3
**Contribution:** 4
**Rating:** 6
**Confidence:** 4

**Summary:**

The authors propose a new LoRA-based approach for customizable generation for diffusion. Their main contributions include proposing a training-free approach and designing new strategies to inject and isolate the concept to ensure the targeted objects are generated without interference. Their approach has been shown to surpass existing SOTA (e.g., Mix-to-Show, Paint-by-Example) with a higher CLIP score on image preservation and text alignment, as well as the mIoU score with the layout.

**Strengths:**

1. The authors propose useful constraints, including concept enhancement and concept isolation, which is an interesting design for the community and can be seen as the plug-and-play objective for future applications.
2. User study is conducted to bridge the gap between human preference and machine metrics. Their results have provided a huge gap under conditions without further image conditions.
3. The model and approach design illustrations are clear and easy to follow. Also, the visualization for different approaches and designs are well-structured.

**Weaknesses:**

1. The authors state that their approach can deal with the concept vanishing issue in Sec. 3.1 but no quantitative comparison to support this statement. For instance, the author can provide a metric that counts how many predicted boxes are obtained with GroundingDINO and compare it with the GT layout. Otherwise, only visualization cannot provide any useful information on how powerfully the proposed approach can deal with the vanishing issue.
2. The authors propose a new dataset but do not provide the results for the existing one proposed in Mix-of-Show. Further discussion or explanation is needed.

**Questions:**

1. More explanation of $\mathcal{L}_c$ in Eq. 3 is needed. What is the meaning of creating $\mathcal{L}_c$? Its form requires the weight within the concept mask to become larger, but why increasing the weight can restrain the activation for the edge region is not clear. Additionally, why can Gaussian weight restrain the activation in the edge region? Can performing a low-pass filter such as blurring get the same results?
2. Sec. 3.4 for latent re-initialization is hard to follow. What is replacing the layout area $z_t[M_i]$ with the latent area? What is the latent area, and how can we obtain it? Missing the latent area can make the paragraph hard to follow.
3. It is observed that the layout for each concept discussed in the paper does not overlap. Is this necessary for the approach to work? What would the outcomes be if some of the boxes overlapped?

---

### Official Review · Reviewer_Wxui · 2024-11-01

**Soundness:** 3
**Presentation:** 3
**Contribution:** 2
**Rating:** 3
**Confidence:** 5

**Summary:**

The paper introduces a training-free model for integrating multiple LoRAs called LoRA-Composer. From given box layouts, a global prompt and local prompts, the proposed method addresses the concept vanishing and confusion issues in multi-concept customization by proposing Concept Injection Constraints and Concept Isolation Constraints, respectively. Concept Injection Constraints modify the cross-attention layers in the U-Net to perform Region-Aware LoRA Injection and Concept Enhancement Constraint, which refine cross-attention maps using Gaussian weighting and adopt a strategy to obtain box-spread attention values. Meanwhile, Concept Isolation Constraints focus on self-attention layers to limit the interaction between queries within a specific concept region and those in other concept regions.
The authors also propose latent re-initialization to obtain better prior latent values for the generation process. LoRA-Composer achieves a notable enhancement compared to standard baselines.

**Strengths:**

- The paper identifies and tackles significant challenges in the multi-concept customization task, which are concept vanishing and concept confusion, by examining the cross-attention and self-attention layers within the U-Net of Stable Diffusion.
- The motivations for the contributions are explained well with informative figures.
- Extensive experiments and ablation studies are conducted to showcase the capability of the proposed method.
- LoRA-Composer can produce visual stunning multi-concept outputs in a training-free manner and does not require the image-based conditions like canny edge or pose estimations. It could potentially have wide applicability across several applications.

**Weaknesses:**

- The novelty of the proposed method is not enough for ICLR:
     + Some contributions should be clarified as either "inspired by existing work to develop" or simply "adopted," in order to emphasize the novelty of the paper:
            . Region-Aware LoRA Injection: Similar to Regionally Controllable Sampling in Mix-of-Show [1]
            . Gaussian weighting in Concept Enhancement Constraints: Similar to the method proposed in BoxDiff [2], with Gaussian weighting from Attend-and-Excite [3].
    + For Region Perceptual Restriction, the idea of minimizing interaction between queries of the foreground and background areas in self-attention is quite popular in existing work related to attention manipulation, such as Attention Refocusing [4].
- The writing in some parts is quite ambiguous:
    + Region-Aware LoRA Injection at line 200: After obtaining h_i in equation (2) at line 215, what do we do next?
    + L_c loss in equation (3) at line 240: What is it? It suddenly appears there without any explanation.
    + Concept Region Mask in Line 270: What do we use it for?
- The prompts used for qualitative evaluation should be mentioned (Figure 5, Figure 6)

[1] Gu, Yuchao, et al. "Mix-of-show: Decentralized low-rank adaptation for multi-concept customization of diffusion models." NIPS 2024
[2] Xie, Jinheng, et al. "Boxdiff: Text-to-image synthesis with training-free box-constrained diffusion." ICCV 2023
[3] Chefer, Hila, et al. "Attend-and-excite: Attention-based semantic guidance for text-to-image diffusion models." ACM Transactions on Graphics (TOG) 2023
[4] Phung, Quynh, Songwei Ge, and Jia-Bin Huang. "Grounded text-to-image synthesis with attention refocusing." CVPR 2024

**Questions:**

- The authors claim that concept injection constraints effectively avoid concept missing (at line 264), but Figure 6(d) still has that issue. So, does the concept isolation constraints (CI) also contribute to the mitigation of concept missing?

- How does the value of k in topk(.) reduce function in the loss components (equation (3) and (7)) affect the results? For example, using larger k might lead to larger generated concepts?

- In scenarios with overlapping box layouts, such as “A [v1] person hugs a [v2] dog,” how effectively does LoRA-Composer perform? It appears that the calculations in these situations may result in many artifacts in the outputs.

- There's a minor analysis point that I think should be clarified. In my view, the Gradient Fusion optimization combined with ED-LoRA introduced in Mix-of-Show [1] is not the primary factor reducing concept identities when generating multi-concept images (e.g., prompts containing multiple concept tokens like “A [v1] man and [v2] woman”). Rather, it's more closely tied to the "incorrect behavior" in the cross-attention and self-attention modules that you are aiming to address. This suggests that the LoRA-Composer method could also be applied to Mix-of-Show or other methods using Gradient Fusion.

[1] Gu, Yuchao, et al. "Mix-of-show: Decentralized low-rank adaptation for multi-concept customization of diffusion models." Advances in Neural Information Processing Systems 36 (2024).

---

### Official Review · Reviewer_6Zmn · 2024-11-03

**Soundness:** 3
**Presentation:** 3
**Contribution:** 3
**Rating:** 6
**Confidence:** 3

**Summary:**

The paper presents LoRA-Composer, a training-free framework designed to manage multi-concept image customization using Low-Rank Adaptations (LoRAs) with layout and textual prompts. LoRA-Composer addresses two key challenges in multi-concept customization: concept vanishing (loss of intended concepts) and concept confusion (misattribution of characteristics between subjects). Key features include concept injection constraints, concept isolation constraints, and latent re-initialization for spatial focus. Experimental results show LoRA-Composer outperforms existing methods in qualitative and quantitative metrics.

**Strengths:**

1. The paper's technical approach appears well-founded. The concept isolation and injection constraints effectively reduce concept vanishing and confusion, supporting the paper's claims of improved performance in multi-concept generation. The latent re-initialization technique also adds rigor, ensuring spatially accurate representation of concepts.
2. The paper is clear and logically structured, guiding readers through the model's design, methodology, and experimental evaluation. Visual examples illustrate improvements over other models.
3. The proposed LoRA Composer is an innovative solution, and the selected baseline should be the latest. In comparison, the model performance of this paper is outstanding, and there are abundant comparative and ablation experiments.

**Weaknesses:**

1. Despite being training-free, the model’s architecture (especially concept isolation and injection constraints) is relatively complex and might limit ease of implementation.
2. Evaluation Scope: The method is tested on select datasets, including COCO, FFHQ, and CelebA-HQ， featuring anime and realistic styles. Testing on broader datasets could enhance its robustness claims.
3. A discussion should be added on whether this method is easy to extend, whether it is applicable to various variants of stable diffusion, and it is not yet clear which version of Stable Diffusion is used in this paper.
4. There seem to be some defects in the figure drawing in the article, such as the arrow pointing to the text encoder in Fig. 2, and there is also a lack of explanation for the data flow related to Fig. 2.

Although the paper has some shortcomings, its overall innovation and the integrity of the experiments are good.

**Questions:**

1. What would be the performance of LoRA-Composer when applied to datasets that exhibit more complex interactions among subjects?
2. Would the fine-tuning of layers beyond the U-Net architecture lead to further enhancements in the preservation of concepts?
3. If the background inherently includes elements of the foreground, would this affect the effectiveness?
4. Would the presence of overlapping layout boxes influence the outcome?
5. Are there any errors in Fig.3a and Fig.3b? It seems that m1-v1 and m2-v2 do not match.
6. For two similar foreground concepts, such as people who look very similar, is there a possibility of concept confusion?

---

### Official Review · Reviewer_ReMW · 2024-11-04

**Soundness:** 1
**Presentation:** 2
**Contribution:** 1
**Rating:** 3
**Confidence:** 4

**Summary:**

The paper presents a modified LORA-based multiple-concept generation model. By introducing three loss functions, the phenomenon of concept vanishing and confusion are somewhat suppressed.

**Strengths:**

The ideas are clearly presented.

**Weaknesses:**

The paper was prepared carelessly. First, the paper exceeds the length limit. Second, the authors claim that the proposed module is training-free. However, the main part is three loss functions. Third, the visualization results are poor. For example in Figure 5, the persons are pasted to the background, and the results look unreal. I think the results of mix-of-show method are far better.

**Questions:**

How to address the issue of adding restrictions on features that significantly reduce the authenticity of generated results?

**Details Of Ethics Concerns:**

N.A.

---

### Note · Authors · 2024-11-15

I have read and agree with the venue's withdrawal policy on behalf of myself and my co-authors.